# Internal Structure, Reliability and Invariance across Gender Using the Multidimensional School Climate Scale PACE-33

**DOI:** 10.3390/ijerph17134800

**Published:** 2020-07-03

**Authors:** Naiara Escalante Mateos, Eider Goñi Palacios, Arantza Fernández-Zabala, Iratxe Antonio-Agirre

**Affiliations:** Department of Developmental and Educational Psychology, Education and Sport Faculty of Vitoria-Gasteiz, University of the Basque Country (UPV/EHU), 01006 Vitoria-Gasteiz, Spain; eider.goni@ehu.eus (E.G.P.); arantza.fernandez@ehu.eus (A.F.-Z.); iratxe.antonio@ehu.eus (I.A.-A.)

**Keywords:** school climate, self-report, internal structure, reliability, invariance

## Abstract

The school climate is one of the keys to the efficiency, quality and improvement of the school. For this reason, numerous studies have highlighted the importance of evaluating this construct. However, there is still no measure in Spanish that evaluates the most relevant aspects of school climate in a valid and reliable way. This paper has two main objectives that try to overcome this limitation: (1) to analyse the internal structure and internal consistency reliability of the Students’ Perception of School Climate scale (escala Percepción del Alumnado sobre el Clima Escolar—PACE-33); and (2) examine its measurement and structural invariance across men and women. This study involved 1180 adolescents (mean age, M_age_ = 15.37 years; standard deviation, SD = 1.90) selected by means of incidental sampling. The results obtained show that, among the models tested, the one corresponding to nine correlated factors shows the best adjustment to the data; furthermore, they note that it presents adequate internal consistency indices. The results also prove that this model is equivalent in men and women. These data support that the PACE-33 is a valid and reliable measure to assess the adolescents’ perception of the main aspects of the school climate.

## 1. Introduction

School climate is considered an important indicator of educational quality [1] and several authors [2,3,4] argue that a positive climate is associated with satisfactory results at both a cognitive—e.g., motivation, academic satisfaction, academic success—and affective-emotional level—e.g., general psychological wellbeing, satisfaction with life and self-esteem. Although this evidence has been recognised for more than a century, over the last twenty years, the scientific-educational community has evinced a growing interest in exploring this construct in greater detail. The research has focused mainly on identifying the factors which foster a positive school climate, or in other words, a climate which is conducive to effective learning and promotes individual wellbeing and personal growth [5,6]. School climate has also often been analysed within the framework of school improvement and reform programmes focused on effective conflict resolution, school effectiveness and interpersonal relationships [7]. However, although numerous studies have explored this construct, finding the right approach remains difficult, since school climate is complex and multifaceted, and has been analysed from many different theoretical and methodological perspectives [8].

There are currently many definitions of school climate, some of which differ greatly [9]. However, although there is no single, agreed-upon definition of the concept, the majority of studies today recognise that proposed by Cohen et al. [10], who argue that school climate refers to the quality and nature of school life. These authors claim that the construct is based on subjects’ experiences at school and reflects said school’s values, aims, rules, interpersonal relationships, habits, teaching and learning practices and organisational structures.

Since school climate has been conceptualised in so many different ways, there is no consistent theoretical approach underpinning the different measurements carried out, and no consensus regarding its different dimensions [1,9]. Although a wide variety of different dimensions have been analysed when studying the construct, some key authors and institutions working in the field of school climate in the English-speaking world [10,11] argue that it comprises four general domains—safety, relationships, environmental-structural aspects and teaching-learning—which are, in turn, divided into different dimensions.

Another aspect which makes the study of school climate difficult is the abundance and disparity of available assessment measures. An updated review of 56 of these instruments highlighted the fact that there is still no valid, reliable measure in Spanish for operationalising adolescent students’ view of the most important dimensions of school climate. Specifically, Escalante et al. [12] found that many existing measures had been designed more than 20 years ago, and had not been psychometrically validated. They also found that, in the vast majority of cases, the theory underpinning the measures was not stated, and that most had been developed in the United States of America, meaning that they had been designed and tested in a country where the sociocultural characteristics and education system are very different from in Spain. Moreover, some studies analysed other variables, such as academic satisfaction [13] or belonging or sense of connection [14] as dimensions of school climate, based on a theoretical conception of the construct that is very different from that which is widely accepted today. Finally, most of the measures available in Spanish principally encompass only those dimensions pertaining to rules and interpersonal relationships, and fail to take into account others, which are equally relevant to the creation and maintenance of a positive school climate. In order to try to solve these limitations, a new instrument called Students’ Perception of School Climate scale (escala Percepción del Alumnado sobre el Clima Escolar—PACE-33) was designed, and its psychometric properties were analysed [15,16]. However, the validation process of this instrument is unfinished, as its factor structure has not yet been corroborated. The PACE-33 is based on the four general domains of school climate—safety, relationships, environmental-structural aspects and teaching-learning—which have the greatest theoretical support [10,11], and are evaluated through specific aspects that have been most included over the years, in different instruments designed to evaluate this construct [15,16].

In addition to the shortcomings outlined above, few studies exist which examine the invariance of the measures developed to assess this construct over the last 15 years. Only 16 studies were found which analyse the invariance of the measures in terms of the sex, ethnic group, academic level, nationality or sexual orientation of the sample, with most focusing solely on measurement invariance. Moreover, even here, only one study examined all four types of invariance—i.e., configural, metric, scalar and strict—which together make up measurement invariance [17]. The majority of studies confirm configural, metric and scalar invariance, either fully [5,18,19,20,21] or partially [22,23,24,25,26,27,28]. Others, however, confirm partial configural, metric and strict invariance [29], or only configural invariance [30]. Moreover, there are a few cases in which, in addition to analysing measurement invariance, the studies also examine the structural invariance of the instruments. For instance, Johnson et al. [29] explored the invariance of the regression coefficients of variances and the residuals of the latent factors; Muñoz et al. [31] confirmed the invariance of the variances and covariances among latent factors; and You et al. [28] partially confirmed the invariance among the mean differences of the latent factors.

Another aspect to be highlighted in the study of school climate is that it has been shown that men and women do not perceive all aspects of the school climate in the same way. Some research states that women perceive the support they receive from their peers more positively than men [23,32], and others argue that men perceive school less positively, and report, in general, more unfavourable perceptions of school climate [33]. Therefore, in order to determine in which specific aspects men and women differ in their perception of the school climate, it is necessary to have instruments that ensure that the most relevant aspects of the school climate are measured in a valid and invariant manner in both sexes.

In light of the above, the present study has two main aims: (1) to analyse the internal structure and internal consistency reliability of the Students’ Perception of School Climate scale (escala Percepción del Alumnado sobre el Clima Escolar—PACE-33); and (2) to examine the measurement and structural invariance of this same scale across men and women.

## 2. Materials and Methods

### 2.1. Participants

The sample initially comprised 1167 students from 11 schools: 8 high schools from the Autonomous Community of the Basque Country, 6 public and 2 semi-private (semi-private schools are private schools which received some government funding), 2 public high schools from Navarre; and 1 public high school from the Principality of Asturias. All cases in which less than 10% of the items had been rated were eliminated, along with those giving inconsistent answers and those with an atypical response pattern, resulting in a final sample of 1138 students from compulsory secondary education (known as ESO in Spain) and the Spanish Baccalaureate (A-level equivalent). Participants were aged between 12 and 20 years (mean age, M_age_ = 15.37 years; standard deviation, SD = 1.90), and 590 (51.85%) were men and 548 (48.15%) women. Of the total, 423 (37.2%) were in the first two years of ESO at the time of the study, 420 (36.9%) were in the final two years of ESO, and 295 (25.9%) were studying for the Spanish Baccalaureate.

### 2.2. Measurement Instrument

Perceived school climate was assessed using the PACE-33 [16]. This instrument comprises 33 items, 29 of which are direct, and 4 inverse (Appendix A, Table A1 and Table A2). The items are grouped into nine subscales (Table 1): (1) physical safety; (2) rules; (3) student-teacher relationships; (4) peer relationships; (5) group cohesion; (6) environmental-structural aspects; (7) teachers’ ability to motivate; (8) teachers’ expectations and (9) methodological resources. Participants respond on a 5-point Likert-type scale, ranging from 1 (totally disagree) to 5 (totally agree).

### 2.3. Procedure

First, 54 schools were selected by means of a purposive sampling technique based on availability. Of these, 11 agreed to participate in the research project during the 2018–2019 and 2019–2020 academic years. After explaining the aim of the study and guaranteeing the confidentiality and voluntary nature of the test, informed consent forms were signed by the school management teams and participating students and their legal guardians. The study was conducted in compliance with the Declaration of Helsinki, and the protocol was approved by the University of the Basque Country’s Ethics Committee for Research Relating to Humans (CEISH-UPV/EHU) (M10_2018_256). To guarantee uniform completion, the PACE-33 was administered simultaneously to all students in each class, and to eliminate any threat to the validity of the results, the anonymous and voluntary nature of student participation was ensured. Although the administrative team was aware of the objectives of this study, the participating students were only made known in a general way that their collaboration was being requested in a study aimed at improving the current situation of the educational system, and optimizing the psychological health and quality of life of adolescents.

### 2.4. Analysis

The statistical program SPSS v.25 (IBM Corporation, Armonk, NY, USA) [34] was used to impute missing values using the linear trend point estimation method, and to identify those cases with atypical values using the anomaly detection procedure. EQS v.6.2 (Multivariate Software, Encino, CA, USA) [35] was used to estimate Mardia’s standardised coefficient of kurtosis, as well as during the subsequent multivariate analyses. The Bentler–Weeks notation system was also used during these analyses.

The study was conducted in accordance with the two established aims. Firstly, to analyse the internal structure of the PACE-33, 9 models representing alternative, theoretically plausible conceptualisations of perceived school climate were compared. To determine the fit of the different models, a confirmatory factor analysis (CFA) was conducted for each one, using the robust maximum likelihood method, since Mardia’s standardised coefficient (85.63) indicated a non-normal multivariate distribution of the data [36]. The fit of the models to the data was assessed using the following robust indices: Satorra–Bentler Chi-squared (SBχ^2^); Satorra–Bentler Chi-squared/degrees of freedom ratio (SBχ^2^/*df*); comparative fit index (CFI), non-normed fit index (NNFI) and normed fit index (NFI), in which values of over 0.90 are deemed acceptable [37]; root mean square error of approximation (RMSEA) and its respective 90% confidence interval, in which values of under 0.06 indicate optimum fit [37]; and the Akaike information criterion (AIC) [38], and the consistent Akaike information criterion (CAIC) [39].

The AIC and CAIC values were used to check for significant differences between the various models tested, since they are valid for assessing the global fit of both nested and non-nested models in a single data set [40]. When the differences are greater than 10, the model with the lowest values in these indices is considered the one with the best fit [41].

Next, the reliability of the scale was tested using the Cronbach’s alpha tau-equivalent measure (α) and McDonald’s omega coefficient of composite reliability (ω) [42], subsequently renamed omega total (ω^T^) by Revelle and Zinbarg [43], as well as Raykov’s rho (ρ) coefficient [44]. Values of over 0.70 are considered acceptable in these three indices [45]. The average variance extracted (AVE) was also estimated as an indicator of the validity of the scale’s internal structure. In this last case, scores equal to or above 0.50 are recommended [45].

Finally, to achieve the study’s second aim, the equivalence of the internal structure of the PACE-33 across men and women was analysed, by studying both the scale’s measurement invariance and its structural invariance [46]. To this end, baseline models were established separately in both groups, making sure the CFAs had acceptable fit indices. Next, the models described below were analysed globally, hierarchically and sequentially for both men and women, imposing accumulative constraints in the successive multigroup CFAs (MG-CFAs).

To determine the measurement invariance of the PACE-33, the next step was to analyse the degree to which the scale’s measurement parameters were equivalent in men and women [47]. To fully assess measurement invariance, however, it is necessary to consecutively analyse configural invariance (M_1_); metric or weak invariance (M_2_); scalar or strong invariance (M_3_); and strict or residual invariance (M_4_) [37]. Firstly, in relation to configural invariance, it was tested whether the number of factors and their composition were the same in both men and women. To this end, the previously established baseline models were estimated simultaneously for both groups, without imposing any constraints on their parameters. Next, to analyse metric invariance, it was tested whether the factor loadings between each item and their corresponding factors were the same in both groups, freely estimating these parameters for men and using them as references, constraining the calculations for women to be the same. Scalar invariance was tested by checking whether the ordinates at the origin of each regression line (i.e., the intercepts) making up the model being tested were the same in both groups. To this end, new constraints had to be added to existing ones in the female group: factor loadings and intercepts were constrained to be the same as those freely estimated for men. Finally, to calculate strict invariance, factor loadings, intercepts and error variance-covariances were freely estimated for men and once again constrained for women (to be the same as those estimated for women).

The structural invariance of the PACE-33 was estimated by analysing the relationships between latent factors. Among the different subtypes of invariance, the covariance invariance (M_5_) and the mean difference invariance (M_6_) of the latent factors were analysed in this study. These analyses are particularly interesting here, because the study of covariance invariance between latent factors enables empirical evidence to be gathered, regarding how the different dimensions which explain adolescent students’ perceptions of school climate are distributed and conceptually related to one another among men and women. For its part, the invariance of the differences between latent means is a prerequisite for guaranteeing that the latent factor means are comparable across the two sexes. As regards the constraints imposed on model M_5_, the equality of covariances between latent factors was added to those constraints previously established for women in the study of measurement invariance. In relation to the invariance of latent mean differences (i.e., M_6_), all the constraints imposed to date were maintained, and differences were estimated indirectly using the procedure described by Byrne [48]: instead of imposing new constraints, the means of the latent factors were fixed at zero among men, and freely estimated for women. Thus, the male group was used as a reference for comparing the latent means calculated for women. Possible significant differences between men and women in latent factor means were determined using *z* > ± 1.96 (*p* < 0.05). Cohen’s *d* [49] was used to assess the magnitude of these differences, and their effect size was determined using the system proposed by the same author (*d*_Cohen_ = 0.200 small; *d*_Cohen_ = 0.500 medium; *d*_Cohen_ = 0.800 large).

The invariance of the PACE-33 was established by comparing the fit of each model with that of the previous one. If the fit of one model did not worsen significantly when new constraints were added to those applied to the previous one, this was taken to indicate that the constraints can be sustained, and that the parameters on which they are imposed operate in the same way in (in this case) men and women. Given that all the models tested were nested, the differences between the CFI and RMSEA goodness-of-fit indices were used to determine whether they were invariant (i.e., ΔCFI ≤ 0.01 and ΔRMSEA ≤ 0.015) [50].

## 3. Results

The results are presented in two sections, each one corresponding to one of the two aims established in the study. The first section outlines the results obtained in relation to the internal structure and reliability of the PACE-33, while the second section presents the results linked to the invariance of this instrument.

### 3.1. Internal Structure and Reliability of the PACE-33

This section outlines the results obtained in relation to the internal structure of the PACE-33, followed by those pertaining to its reliability.

#### 3.1.1. Comparison of Alternative Models of Perceived School Climate

In order to determine whether the internal structure of the PACE-33 corresponds to the theoretical construct it was designed to measure, nine models representing alternative, theoretically plausible conceptualisations of the perceived school climate were compared. Figure 1 presents these nine models consecutively, in order of complexity, from simplest to most complex. Their conceptualisation is explained below.

Model M_A_ represents a unidimensional conceptualisation of perceived school climate, in global terms. Although the authors of previous studies coincide in rejecting a unidimensional theoretical conceptualisation of this construct [51], it is best to determine the goodness-of-fit of single-factor measurement models prior to assessing other, structurally more complex ones [37].

M_B_ represents four correlated oblique factors—safety, relationships, environmental-structural aspects and teaching-learning—which correspond to the four domains established by the National School Climate Center [11] and Cohen et al. [10]

M_C_ represents nine correlated oblique factors—physical safety; rules; student-teacher relationships; peer relationships; group cohesion; environmental-structural aspects; teachers’ ability to motivate; teachers’ expectations; and methodological resources—which represent the most important dimensions of perceived school climate [15].

M_D_ has a two-level higher-order structure, in which factors corresponding to the four general domains previously described in M_B_ are grouped into a single factor linked to perceived general school climate. According to this model, it is adolescent students’ global perceptions of the climate at their school that are responsible for the joint variation observed in scores for the factors safety, relationships, environmental-structural aspects and teaching-learning.

Similarly, Model M_E_ also postulates a two-level higher-order structure, although in this case with nine (rather than four) first-order factors. These factors correspond to the most important specific dimensions of perceived school climate, as described previously in M_C_, grouped into a single higher-order factor—perceived general school climate—which is responsible for the variance observed in the first-order factors.

M_F_ posits a three-level structure with nine first-order factors corresponding to the principal dimensions of perceived school climate (as in M_C_), of which two—teachers’ ability to motivate and methodological resources—are associated with the higher-order factor motivational resources. The remaining seven first-order factors and the higher-order factor motivational resources are in turn grouped into a general factor corresponding to perceived school climate. This structure was tested in order to clarify whether the close association observed previously between the factors teachers’ ability to motivate and methodological resources [16] is due to the existence of a higher-order factor which encompasses them.

M_G_ also posits a three-level higher-order structure. The factors physical safety and rules are grouped into the higher-order factor safety; student-teacher relationships, peer relationships and group cohesion are grouped into the higher-order factor relationships; environmental-structural aspects is not included in any higher-order factor; and teachers’ ability to motivate, teachers’ expectations and methodological resources are grouped into the higher-order factor teaching-learning. The higher-order factors safety, relationships, environmental-structural aspects and teaching-learning are in turn grouped into a general factor corresponding to perceived school climate.

M_H_ comprises a bifactorial structure that can be considered an alternative to the second-order models. The main differences between these two structures is that, in second-order models, the general factor reflects the common variance of all the lower-order latent factors, while in bifactorial models, the general factor reflects only part of the common variance observed between the items on the scale [52]. Specifically, M_H_ represents a structure in which each of the PACE-33 items saturates on a single general factor—perceived school climate—as well as on the specific factor for which it was designed and which corresponds to one of the established dimensions. These factors are orthogonal in relation to both each other and the general factor. In other words, each item is directly affected by both the general factor and the specific factor to which it belongs.

The final model, M_I_, also represents a bifactorial structure of perceived school climate. In this case, however, it is the four factors corresponding to the domains of school climate defined previously in M_B_ that explain part of the common variance observed in the items, with the rest of the common variance being explained by the nine factors defined in M_C_.

Table 2 shows the different robust goodness-of-fit indices obtained for the nine alternative models tested.

Three of the models tested, specifically M_A_, M_B_ and M_D_, had a particularly poor fit. In these models, the SBχ^2^/*df* ratio was not around 2, as recommended by Tabachnick and Fidell [53]; the NFI, NNFI and CFI indices did not reach the 0.90 cut-off point, and the RMSEA had a value of over 0.06. They cannot, therefore, be considered to have an acceptable fit. The rest of the models tested, however, were found to have values above the cut-off points established to indicate acceptability. Among these models, M_C_ and M_H_ were the two with the best fit, although M_C_ was the model with the lowest values in the AIC and CAIC information criteria. Since the difference between this model and the rest in these indices was greater than 10, it can be considered the most parsimonious, and therefore the one with the best fit [41]. Thus, M_C_—the model comprising nine correlated factors—was the one with the greatest empirical support, and was therefore the one used in all subsequent analyses.

In this model, the frequency distribution of the standardised residuals revealed that 96.26% of the residuals had central values of between −0.1 and 0.1. As regards the non-standardised coefficients and the standard errors, all were found to have reasonable and statistically significant values. In relation to the standardised regression coefficients, all items had loadings of over γ > 0.40, and the regression coefficients (Table 3) between the items and the factors ranged between 0.512 and 0.908. As regards the relationships between latent factors (Table 3), the significant covariances oscillated between 0.047 and 0.629, with peer relationships and teachers’ expectations (F4-F8) being those with the weakest association, and student-teacher relationships and teachers’ ability to motivate (F3-F7) being those with the strongest. The Wald test revealed that physical safety had no significant relationship with rules (F1-F2), peer relationships (F1-F4) or teachers’ ability to motivate (F1-F7). Furthermore, it was also observed that the relationship between teachers’ expectations and methodological resources (F8-F9) was not significant.

#### 3.1.2. Internal Consistency Reliability and AVE of the PACE-33

Table 4 shows the different internal consistency reliability coefficients for the PACE-33 subscales. The AVE is also given as an indicator of the validity of their internal structure.

All subscales were found to have adequate values of over 0.70 [45]. These values were between 0.712 and 0.918 for Cronbach’s alpha; between 0.722 and 0.918 for McDonald’s omega; and between 0.725 and 0.918 for Raykov’s rho.

With regards to the AVE, all subscales had adequate values of over 0.50 [45], with the exception of the subscale environmental-structural aspects, which had a lower value (AVE = 0.401).

### 3.2. Analysis of the Invariance of the PACE-33 Scale by Sex

Both the measurement invariance and the structural invariance of the PACE-33 were analysed in this study. Firstly, the configural, metric, scalar and strict invariances were analysed, in order to calculate measurement invariance. Next, the equivalence of the covariances between the latent factors was tested, as well as the differences in their latent means, in order to calculate structural invariance.

Firstly, the model with nine correlated factors was tested separately for men and women (Table 5), in order to determine whether it had an acceptable fit in both groups. The nine correlated factor model was established as the baseline model for each separate group (M_0_ men and M_0_ women), in accordance.

The item-factor factor loadings were also found to have adequate values in both groups (Table 6).

Nevertheless, the Wald test indicated that some of the relationships between the latent factors were not significant (Table 7). Specifically, for both men and women, non-significant relationships were found between physical safety and rules (F1–F2); physical safety and peer relationships (F1–F4); and physical safety and teachers’ ability to motivate (F1–F7). Moreover, for men, the results indicated that there was a non-significant association, either between physical safety and teachers’ expectations (F1–F8); physical safety and methodological resources (F1–F9); and rules and peer relations (F2–F4). For women, other relationships were found which were not significant: physical safety and student-teacher relationships (F1–F3); peer relationships and teachers’ expectations (F4–F8), and teachers’ expectations and methodological resources (F8–F9). Variations between the baseline models of the two groups are permitted, providing that the number of items and factors is maintained and the relationships are similar [48]. In this case, the covariances, which were not significant, were estimated freely in both groups.

Once the baseline model had been established for both sexes, measurement invariance was explored in a series of successive analyses designed to determine configural, metric, scalar and strict invariance. As shown in Table 5, no significant differences were observed between the models (ΔCFI_MIN-MAX_ = 000–0.002; ΔRMSEA_MIN-MAX_ = 000–0.001), thus indicating that the successive constraints imposed may be maintained. The results therefore revealed that the arrangement of the different items and latent factors (M_1_), item-factor factor loadings (M_2_), item intercepts (M_3_) and variance in item errors (M_4_) are the same in both groups (men and women).

As regards to the analysis of the structural invariance of the PACE-33, as shown again in Table 5, no significant differences were found when M_5_, the model with the added constraint of covariance equality between latent factors, was compared with the last model tested for measurement invariance (M_4_). Therefore, it can be concluded that both groups have a similar pattern of relationship intensity between latent factors. Furthermore, no significant differences were found between the latent factor means, with the exception of the factor physical safety (F1), which was observed to have significantly lower scores (*z* = −2.52, *p* < 0.05) among women. The estimation value suggests that, among women, safety values were −0.150 units lower than among men, although the effect size was small (*d*_Cohen_ = 0.149).

## 4. Discussion

This study analyses the internal structure, reliability, measurement invariance and structural invariance of the PACE-33 in a sample of Spanish adolescents. The study aims to redress the lack of valid, reliable instruments in Spanish for measuring adolescent students’ perceptions of the most relevant dimensions of school climate, which is one of the most important variables in school efficacy, quality and improvement [1,54]. This paper offers a valid and reliable instrument for assessing this construct. The results indicate that the PACE-33 offers an effective and precise measurement of factors linked to perceptions of physical safety, rules, teacher-student relations, peer relations, group cohesion, environmental-structural aspects, teachers’ ability to motivate students, teachers’ expectations and methodological resources among students aged between 12 and 18 years. The study not only provides empirical evidence of the structure of the perceived school climate, it also clearly reveals that, with the exception of the physical safety subscale, there are no differences between the way men and women perceive this construct. The results are discussed in more detail below.

In relation to the first aim, which was to analyse the internal structure and reliability of the PACE-33, the results reveal that the model comprising nine correlated factors had the best fit of all those tested, and is therefore the one that best reproduces the observed data. This structure reflects the arrangement of the dimensions that together make up school climate, and is consistent with that reported by other authors, such as Hung et al. [55], Muñoz et al. [31] and Yang et al. [27], who, despite validating instruments comprising subscales different from those of the PACE-33, nevertheless identified a structure made up of different correlated factors. Similarly, Quijada et al. [1] also confirmed the existence of a correlated factor structure in relation to teachers’ perceptions of school climate.

The data obtained in the study suggest that some of the relationships observed among the nine factors are not significant. Specifically, physical safety had no significant relationship with rules, peer relations or teachers’ ability to motivate; and no significant association was found either between teachers’ expectations and methodological resources. The fact that physical safety was not related to some of the other factors may be due to the drafting and content of the items, which, unlike the rest of the scale, are written inversely, and describe disruptive behaviour observed among other students which may make individuals feel unsafe at school (e.g., “At my school there are students who threaten and insult others”). The results of the present study are consistent with that reported by Bear et al. [19], who observed that the factor student conduct problems had no statistically significant relationship with two of the other five factors comprising the school climate model presented: teacher-student relations and fairness of rules.

As regards the absence of a significant association between the factors teachers’ expectations and methodological resources, one possible explanation is that they reflect different aspects. The methodological resources factor reflects the specific strategies that students identify as having been used by their teachers in the classroom, while the factor teachers’ expectations reflects what students believe their teachers think of their level of interest and academic performance. In other words, while the former refers to situations that students have actually experienced, the latter refers to their beliefs about their teachers’ expectations of their behaviour. Future research should try to corroborate the absence of any relationship between these two aspects, and reflect on the need to theoretically redefine the dimension physical safety, and find a more adequate way of measuring it. Despite this, however, it is important to note that all the subscales can be used to measure the different theoretical aspects of perceived school climate, regardless of the relationships established among them.

The results of the present study also indicate that all nine subscales of the PACE-33 have adequate internal consistency reliability indices, meaning that they all provide precise measurements of the main dimensions of perceived school climate. This sets the PACE-33 apart from other scales designed to measure this same construct, which do not have adequate coefficients for all their subscales. For example, the physical conditions subscale of the School Social Climate Scale (ECLIS) by Aron et al. [54], and the students support subscale of the instrument developed by Hung et al. [55], have lower values.

Another aspect which attests to the validity of the internal structure of the PACE-33 is the fact that most of the subscales have AVE values of over 0.50, indicating that at least half of the variation observed in their items is due to the latent factor to which they belong [45]. The subscale environmental-structural aspects, however, were found to have a value of under the established cut-off point (AVE = 0.401). Nevertheless, according to the criterion outlined by Bettencourt [56], this value can be considered acceptable, since the composite reliability coefficients of this subscale are reasonable (ω > 0.722 and ρ > 0.724), and the corrected item-subscale correlations are over 0.40.

As regards the study’s second aim, which refers to the measurement and structural invariance of the PACE-33 among men and women, the results obtained indicate that the model itself, relationships between factors and all the latent means are, with the exception of that referring to physical safety, similar in both sexes. This means the following: (1) the arrangement of the factors which make up perceived school climate is similar among men and women and no significant relationships exist in either group between physical safety and rules, physical safety and peer relations, and physical safety and teachers’ ability to motivate; (2) in the case of men, non-significant relationships were also found between physical safety and teachers’ expectations, physical safety and methodological resources, and rules and peer relations; (3) among women, non-significant relationships were also observed between physical safety and student-teacher relations, peer relations and teachers’ expectations, and teachers’ expectations and methodological resources; and (4) men and women are comparable in terms of observable and latent scores, with the exception of the physical safety subscale, in which women’s latent scores were significantly lower than men’s. The differences observed between women and men in the relationships between latent factors, and the significant differences found between their scores in the latent factor physical safety, may be due to the fact that the items which make up this subscale are the only ones written inversely. Another possible explanation may be that these differences are due to the physiological characteristics and/or stereotypes attributed to men and women, which may prompt the two sexes to interpret the theoretical content of said items differently [57,58].

Another aspect worth highlighting is the fact that this study provides evidence of both the measurement and the structural invariance of the PACE-33. Unlike most previous studies, which focus solely on analysing measurement invariance [5,18,19,20,21,22,23,24,25,26,27,28,30], this paper analyses and confirms, in a single study, strict invariance; the invariance of the regression coefficients between latent factors and the invariance of the differences in the majority of latent means, aspects which have hardly been studied at all in previous research [28,29,31].

It should be highlighted that, in addition to having adequate psychometric properties, the PACE-33 is a brief scale that contains only 33 items measuring nine factors. Brevity is a particularly important quality in instruments designed to be used in the educational field, in which completion time is limited; there is a risk of students becoming fatigued and many different instruments are often administered together [59].

The present study is not without its limitations. Firstly, when studying the PACE-33, the results obtained here could only be compared with those reported previously in qualitative terms, since no other school climate measurement instrument assesses the same aspects as those which make up this scale. Moreover, although the sample was broad and adequate for the analyses carried out, since it only comprised students from three of Spain’s seventeen Autonomous Communities, future research may wish to corroborate the results obtained here with students from other regions of the country, as well as from other cultural contexts. Similarly, since the study only analysed the invariance of the PACE-33 instrument in relation to sex, it would be interesting for future research to analyse its invariance in relation to other variables, such as school year or key stage, type of school (semi-private or public), and the size of the school, as well as the socioeconomic status of families. Additionally, since the PACE-33 scale is aimed at knowing only students’ perception of the most relevant aspects of school climate, it would be interesting for future research to adapt and apply this scale to other members of the educational community—teachers, parents and staff—along the lines of those carried out by authors such as Bocchi et al. [60], in order to obtain a more complete view of school climate in each educational centre.

Despite these limitations, however, the study confirms the validity, reliability and invariance (in terms of sex) of the brief scale, and guarantees its suitability for assessing adolescent students’ perceptions of the most important aspects of school climate.

## 5. Conclusions

This study furthers our knowledge of school climate from a twofold psychometric and theoretical perspective. Firstly, it provides a brief, valid and reliable instrument for measuring adolescent students’ perceptions of the most important aspects of school climate—the PACE-33—thereby redressing the observed lack of an instrument with these characteristics suitable for use in the Spanish academic field.

It therefore contributes to improving the measurement of perceived school climate, which in turn, will help teachers and researchers understand more precisely how adolescents view this construct and enable them to identify those aspects which can be improved, in order to foster a positive school climate, work on their development, and provide clearer educational guidelines when addressing them. Secondly, since the scale enables men’s and women’s perceptions of these aspects to be compared, it also aids the design of sex-specific psychosocial interventions at school, as well as the establishment of more comprehensive proposals.

Finally, on a theoretical level, the study helps to delimit the construct by identifying the principal aspects which comprise it—an important step forward in the field of research into the school climate.

## Figures and Tables

**Figure 1 ijerph-17-04800-f001:**
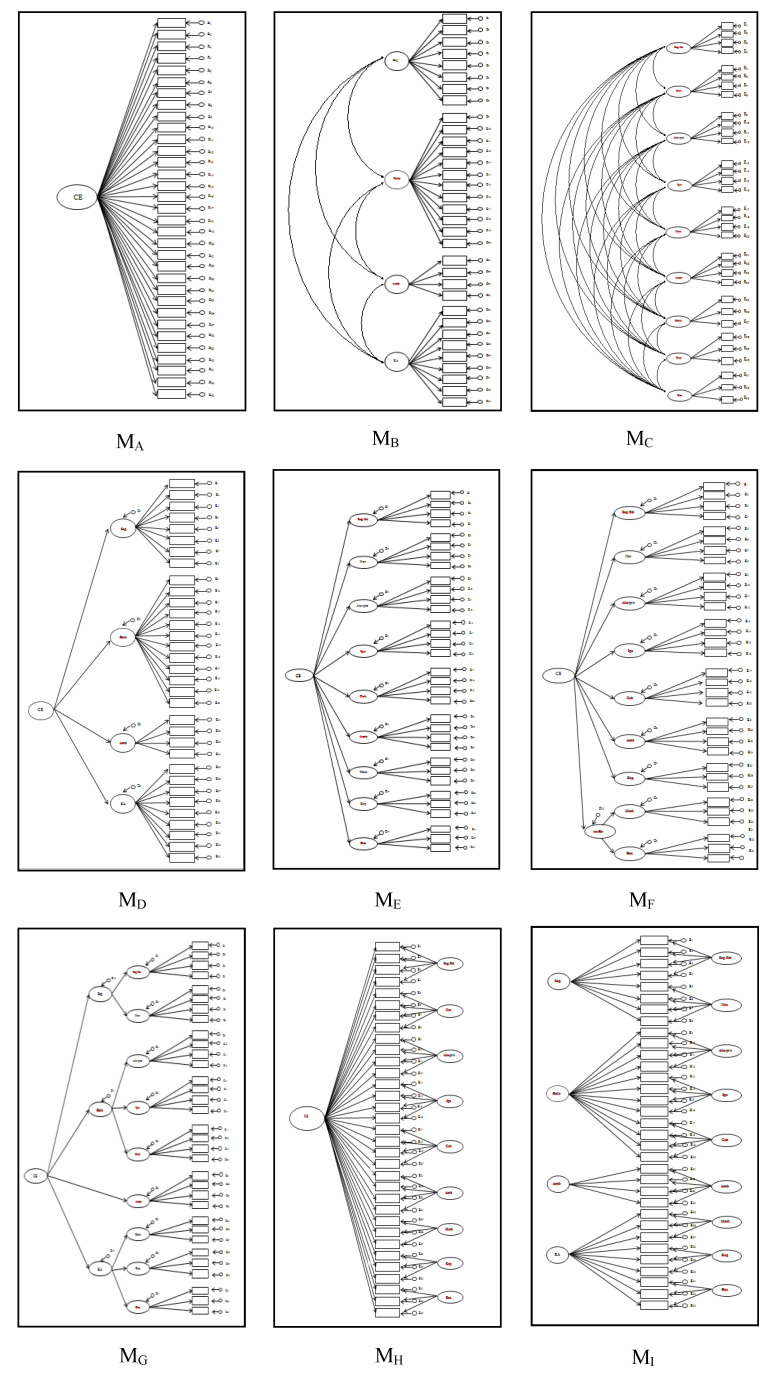
Alternative and theoretically plausible models of school climate.

**Table 1 ijerph-17-04800-t001:** Definition of the PACE-33 subscales.

Subscale	Definition
Physical safety	Respondents’ perception of danger, harm and/or risks at school that make them feel they are not safe there.
Rules	Respondents’ perception of the rules in effect at school—e.g., whether there are rules, how they are communicated.
Student-teacher relationships	Respondents’ impression of the relationship they have with their teachers—e.g., good communication, trust, etc.
Peer relationships	Respondents’ perception of the relationship they have with their peers—e.g., good communication, trust.
Group cohesion	Respondents’ impression of their classmates’ tendency to respect and help each other, and their class group’s tendency to remain united as a cohesive whole.
Environmental-structural aspects	Respondent’s perception of the different environmental-structural aspects of their school, which impact their wellbeing and health—e.g., cleanliness and lighting in the different areas and rooms.
Teachers’ ability to motivate	Respondents’ impression of their teachers’ ability to influence how they act—e.g., to motivate them—encouraging them to work enthusiastically towards achieving their goals and aims.
Teachers’ expectations	Respondents’ impression of their teachers’ educational expectations and interest in the academic success of all their students.
Methodological resources	Respondents’ perception of the methodological resources used by their teachers—e.g., innovative resources.

**Table 2 ijerph-17-04800-t002:** Goodness-of-fit indices for the nine models tested.

Indices	M_A_	M_B_	M_C_	M_D_	M_E_	M_F_	M_G_	M_H_	M_I_
SBχ^2^ (*df*)	10313.19 (495) **	8006.10 (489) **	1093.92 (459) **	8009.42 (491) **	1557.37 (486) **	1516.93 (485) **	1520.46 (483) **	1335.47 (462) **	2024.49 (461) **
SBχ^2^/*df*	20.83	16.37	2.38	16.31	3.21	3.12	3.15	2.89	4.39
NFI	0.395	0.530	0.936	0.530	0.909	0.911	0.911	0.922	0.881
NNFI	0.366	0.509	0.956	0.510	0.930	0.932	0.931	0.940	0.892
CFI	0.405	0.545	0.962	0.545	0.935	0.938	0.937	0.947	0.905
RMSEA _(CI 90%)_	0.132 _(0.130, 0.134)_	0.116 _(0.114, 0.118)_	0.035 _(0.032, 0.038)_	0.116 _(0.114, 0.118)_	0.044 _(0.042, 0.046)_	0.043 _(0.041, 0.046)_	0.043 _(0.041, 0.046)_	0.041 _(0.038, 0.043)_	0.055 _(0.052, 0.057)_
AIC	9323.19	7028.10	175.92	4063.24	585.37	546.93	554.46	411.47	1102.49
CAIC	6334.86	4075.99	−2595.08	7027.42	−2348.63	−2381.03	−2361.43	−2377.64	−1680.58

Note. ** *p* < 0.01; M_A_ = Unidimensional model; M_B_ = Model with four correlated factors; M_C_ = Model with nine correlated factors; M_D_ = Two-level model—four first-order factors and one higher-order factor; M_E_ = Two-level model—nine first-order factors and one higher-order factor; M_F_ = Three-level model—nine first-order factors, two of which are grouped into a second-order factor, and one higher-order factor; M_G_ = Three-level model—nine first-order factors, four second-order factors and one higher-order factor; M_H_ = Two-factor model—nine specific factors and one general factor, all orthogonal; M_I_ = Two-factor model—nine specific factors and four general factors, all orthogonal.

**Table 3 ijerph-17-04800-t003:** Covariances between latent factors and item-factor regression coefficients.

Items	F1	F2	F3	F4	F5	F6	F7	F8	F9
F1	-								
F2	0.029	-							
F3	0.104 **	0.36 **	-						
F4	−0.004	0.074 **	0.137 **	-					
F5	0.206 **	0.091 **	0.181 **	0.271 **	-				
F6	0.118 **	0.211 **	0.348 **	0.130 **	0.188 **	-			
F7	0.046	0.278 **	0.629 **	0.094 **	0.212 **	0.303 **	-		
F8	−0.054 *	0.128 **	0.322 **	0.047 *	0.099 **	0.168 **	0.347 **	-	
F9	0.099 **	0.284 **	0.500 **	0.088 **	0.192 **	0.296 **	0.579 **	0.041	-
PACE01	0.804								
PACE02	0.817								
PACE03	0.637								
PACE04	0.821								
PACE05		0.727							
PACE06		0.766							
PACE07		0.822							
PACE08		0.765							
PACE09			0.804						
PACE10			0.817						
PACE11			0.637						
PACE12			0.821						
PACE13				0.727					
PACE14				0.777					
PACE15				0.783					
PACE16				0.863					
PACE17					0.760				
PACE18					0.845				
PACE19					0.748				
PACE20					0.748				
PACE21						0.628			
PACE22						0.512			
PACE23						0.687			
PACE24						0.689			
PACE25							0.866		
PACE26							0.908		
PACE27							0.891		
PACE28								0.729	
PACE29								0.834	
PACE30								0.870	
PACE31									0.808
PACE32									0.840
PACE33									0.874

Note. * *p* < 0.05; ** *p* < 0.01; F1 = Physical safety; F2 = Rules; F3 = Student-teacher relationships; F4 = Peer relationships; F5 = Group cohesion; F6 = Environmental-structural aspects; F7 = Teachers’ ability to motivate; F8 = Teachers’ expectations; F9 = Methodological resource.

**Table 4 ijerph-17-04800-t004:** Reliability coefficients and average variance extracted (AVE) of the PACE-33 subscales.

Subscale	α	ω	ρ	AVE
Physical safety	0.851	0.852	0.855	0.598
Rules	0.853	0.853	0.854	0.594
Student-teacher relationships	0.855	0.841	0.841	0.596
Peer relationships	0.866	0.867	0.868	0.623
Group cohesion	0.856	0.857	0.858	0.603
Environmental-structural aspects	0.712	0.722	0.725	0.401
Teachers’ ability to motivate	0.918	0.918	0.918	0.789
Teachers’ expectations	0.849	0.851	0.853	0.661
Methodological resources	0.878	0.878	0.879	0.707

Note. α = Cronbach’s alpha; ω = McDonald’s omega; ρ = Raykov’s rho; AVE= Average Variance Extracted.

**Table 5 ijerph-17-04800-t005:** Goodness-of-fit indices and invariance tests between men and women for the PACE-33.

Model	SBχ^2^ (*df*)	SBχ^2^/*df*	NFI	NNFI	CFI	RMSEA _(CI90%)_	AIC	CAIC	Model Comparison		
										∆CFI	∆RMSEA
M_0M_	826.70 (465) **	1.78	0.907	0.951	0.957	0.036 _(0.032, 0.040)_	−100.68	−2570.16			
M_0W_	745.52 (465) **	1.60	0.915	0.961	0.966	0.033 _(0.029, 0.038)_	−177.23	−2612.81			
M_1_	1573.01 (930) **	1.69	0.911	0.956	0.961	0.035 _(0.032, 0.038)_	−286.99	−5901.42			
M_2_	1594.96 (954) **	1.67	0.910	0.957	0.961	0.034 _(0.031, 0.037)_	−313.04	−6072.36	M_1_–M_2_	0.000	0.001
M_3_	1667.35 (987) **	1.69	0.910	0.955	0.961	0.035 _(0.032, 0.038)_	−3060.64	−62650.19	M_2_–M_3_	0.000	0.001
M_4_	1729.91 (1020) **	1.70	0.907	0.954	0.959	0.035 _(0.032, 0.038)_	−3100.09	−64670.85	M_3_–M_4_	0.002	0.000
M_5_	1764.45 (1047) **	1.68	0.905	0.955	0.959	0.035 _(0.032, 0.038)_	−3290.55	−66500.32	M_4_–M_5_	0.000	0.000
M_6_	1756.09 (1039) **	1.69	0.905	0.954	0.959	0.035 _(0.032, 0.038)_	−3210.91	−65940.38	M_5_–M_6_	0.000	0.000

Note. ** *p* < 0.01; CI 90% = Confidence Interval at 90%; M_0M_ = Independent baseline model for men; M_0W_ = Independent baseline model for women; M_1_ = Configural model; M_2_ = Metric model; M_3_ = Scalar model; M_4_ = Strict model; M_5_ = Latent factor covariance model; M_6_ = Model of the differences between the means of the latent factors.

**Table 6 ijerph-17-04800-t006:** Item-factor loadings in the baseline models for men and women.

Items	Item-Factor Loadings	
	M_0M_	M_0W_
PACE01-F1	0.805	0.799
PACE02-F1	0.831	0.799
PACE03-F1	0.683	0.576
PACE04-F1	0.838	0.796
PACE05-F2	0.743	0.715
PACE06-F2	0.757	0.779
PACE07-F2	0.813	0.830
PACE08-F2	0.737	0.794
PACE09-F3	0.755	0.833
PACE10-F3	0.734	0.767
PACE11-F3	0.719	0.720
PACE12-F3	0.765	0.753
PACE13-F4	0.713	0.744
PACE14-F4	0.783	0.776
PACE15-F4	0.756	0.810
PACE16-F4	0.873	0.853
PACE17-F5	0.722	0.799
PACE18-F5	0.839	0.856
PACE19-F5	0.727	0.763
PACE20-F5	0.738	0.765
PACE21-F6	0.657	0.577
PACE22-F6	0.547	0.458
PACE23-F6	0.657	0.734
PACE24-F6	0.656	0.745
PACE25-F7	0.872	0.859
PACE26-F7	0.917	0.897
PACE27-F7	0.892	0.888
PACE28-F8	0.741	0.717
PACE29-F8	0.829	0.840
PACE30-F8	0.860	0.883
PACE31-F9	0.786	0.835
PACE32-F9	0.832	0.850
PACE33-F9	0.860	0.888

Note. M_0M_ = Independent baseline model for men; M_0W_ = Independent baseline model for women; F1 = Physical safety; F2 = Rules; F3 = Student-teacher relationships; F4 = Peer relationships; F5 = Group cohesion; F6 = Environmental-structural aspects; F7 = Teachers’ ability to motivate; F8 = Teachers’ expectations; F9 = Methodological resources.

**Table 7 ijerph-17-04800-t007:** Covariances between latent factors.

Subscale	F1	F2	F3	F4	F5	F6	F7	F8	F9
**F1**	-	0.030	0.082	0.018	0.192 **	0.075 *	0.009	−0.087 **	0.121 **
**F2**	0.032	-	0.338 **	0.099 **	0.107 **	0.189 **	0.228 **	0.082 **	0.290 **
**F3**	0.116 *	0.387 **	-	0.132 **	0.129 **	0.337 **	0.565 **	0.272 **	0.474 **
**F4**	−0.008	0.049	0.139 **	-	0.286 **	0.106 **	0.073 *	0.021	0.088 *
**F5**	0.214 **	0.079 **	0.223 **	0.257 **	-	0.126 **	0.143 **	0.066 *	0.174 **
**F6**	0.149 **	0.217 **	0.344 **	0.146 **	0.239 **	-	0.239 **	0.130 **	0.275 **
**F7**	0.075	0.331 **	0.685 **	0.115 **	0.271 **	0.355 **	-	0.304 **	0.517 **
**F8**	−0.023	0.177 **	0.365 **	0.071 *	0.128 **	0.202 **	0.386 **	-	−0.028
**F9**	0.080	0.280 **	0.524 **	0.089 *	0.210 **	0.303 **	0.639 **	0.107 *	-

Note. * *p* < 0.05; ** *p* < 0.01; F1 = Physical safety; F2 = Rules; F3 = Student-teacher relationships; F4 = Peer relationships; F5 = Group cohesion; F6 = Environmental-structural aspects; F7 = Teachers’ ability to motivate; F8 = Teachers’ expectations; F9 = Methodological resources. The lower diagonal shows the covariances between the latent factors in the baseline model for men; the upper diagonal shows the covariance in the baseline model for women.

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
