# Peer review of "Internal Structure, Reliability and Invariance across Gender Using the Multidimensional School Climate Scale PACE-33"

_ijerph, 2020, doi:10.3390/ijerph17134800_

Round 1
Reviewer 1 Report
The article covers in depth the internal structure of the Students' Perception of School Climate scale PACE-30 and offers interesting results on the Analysis of the Invariance of the PACE-33 Scale by Sex. The introduction provides a good review of the problems of other scales of Social Climate, and reflects the need for studies of this type to be contextualized within a specific country and educational system. However, there is a lack of the theoretical model that underlies the PACEO-30 scale. There is also no reference to the ownership of the test neither to the validation process, information that is relevant to assess the quality of the study.

Author Response
First of all, we would like to express our sincere appreciation as your comment has enabled us to improve the introduction of the article. We have included a new paragraph (lines 69-76) in which we explain the theoretical model that underlies the PACE-33 scale, we detail its authorship and we refer to the two previous articles in which the validation process of this scale can be consulted.
Reviewer 2 Report
I would like to express the encouragement for authors that they decided to analyze such actual topic. The quality of the paper is good. I have no notes, comments.
Author Response
| Thank you very much for your kind words. |
Reviewer 3 Report
The school climate is recognised as a systemic model. therefore it is a limitation to have given questionnaires only to students. Many studies have shown the importance of the parent-teacher relationship and the leadership of the school Head for the school climate. It is clear that only one particular aspect was investigated, namely gender and differently expressed perception. Although the initial objectives do not specify this. Would it have been interesting to explain why such a criterion was chosen, was it supported by previous studies?
At line 118 it is said that the students were not informed about the topic of the research. That would be a questionable point. At least it would be interesting to know how the problem was solved (debriefing?).
There is an Italian SCPQ questionnaire that investigates the school climate, adapted to the 2013 European context (Bocchi, Cavrini, Dozza, Chianese).
Author Response
First of all, we would like to express our sincere gratitude. Your appreciation and comments have enabled us to improve the manuscript and to reflect on various aspects of it.
In the introduction of the manuscript we have introduced a new paragraph (lines 91-98) in which we explain, based on previous studies, why do we choose to analyse the invariance of this scale according to sex. However, as we already recognized in the limitations of this study, it would be convenient for future research to analyse the invariance of the PACE-33 scale based on other variables such as the school year or the size of the educational centre. In the discussion of the article (lines 512-517), we have also added that another of the limitations of this study is that the PACE-33 scale has only been applied to students and we have shown that it would be interesting for future studies to adapt and apply this scale to other members of the educational community, as Bocchi, Cavrini, Dozza, and Chianese (2013) do in their research. Finally, we have specified in more detail what measures we take to try to eliminate any threat to the validity of the results (lines 133-137).